# Super-formable pure magnesium at room temperature

Zhuoran Zeng[1], Jian-Feng Nie[1], Shi-Wei Xu[2], Chris H. J. Davies[3] & Nick Birbilis [1]

Magnesium, the lightest structural metal, is difficult to form at room temperature due to an insufficient number of deformation modes imposed by its hexagonal structure and a strong texture developed during thermomechanical processes. Although appropriate alloying additions can weaken the texture, formability improvement is limited because alloying additions do not fundamentally alter deformation modes. Here we show that magnesium can become super-formable at room temperature without alloying. Despite possessing a strong texture, magnesium can be cold rolled to a strain at least eight times that possible in conventional processing. The resultant cold-rolled sheet can be further formed without cracking due to grain size reduction to the order of one micron and inter-granular mechanisms becoming dominant, rather than the usual slip and twinning. These findings provide a pathway for developing highly formable products from magnesium and other hexagonal metals that are traditionally difficult to form at room temperature.

[1] Department of Materials Science and Engineering, Monash University, Melbourne, Vic 3800, Australia. [2] Automotive Steel Research Institute, Research Institute (R&D Centre), Baoshan Iron & Steel Co., Ltd, Shanghai 201900, China. [3] Department of Mechanical and Aerospace Engineering, Monash University, Melbourne, Vic 3800, Australia. Correspondence and requests for materials should be addressed to J.-F.N. (email: jianfeng.nie@monash.edu) or to N.B. (email: nick.birbilis@monash.edu)

Magnesium is a widely available metal. Comprising 2.7% of the earth's crust it is readily commercially produced from seawater and from its ore with a purity that can exceed 99.8%. It has a density that is 66% of aluminium and 25% of steel. These unique features make magnesium a promising candidate for substituting steel and aluminium alloys for more energy efficient and environmentally friendly applications[1, 2]. For example, each 100 kg reduction in vehicle weight reduces fuel consumption by 0.38 litre per 100 kilometre and $CO_2$ emission by 8.7 g per kilometre[3]. One major barrier to the wide use of magnesium products is their limited formability: magnesium itself is intrinsically difficult to form at room temperature. Deformation modes that are commonly activated in magnesium are slip on the basal plane, prismatic planes and pyramidal planes, as well as twinning[4]. The available slip deformation modes are progressively more difficult to activate and this is compounded by a strong basal texture developed during thermomechanical processing[4, 5]. Twinning is highly dependent on orientation and exhausts after all suitably oriented grains have twinned, usually at around a strain of up to 0.08[6]. As a result, in contrast to the substantial formability of aluminium in cold rolling (recalling that aluminium beverage can walls are around 100 μm thick and aluminium foil is an everyday item), fracture usually occurs when polycrystalline pure magnesium is cold rolled by only ~30% thickness reduction[7]. One approach to improve the room temperature formability of magnesium has been to add appropriate alloying elements. Alloying additions can reduce the stress required to activate more deformation modes and/or weaken the basal texture to allow easier plastic deformation[8–13]. While such efforts have achieved some success in terms of formability improvement, they have not developed any magnesium products that are super-formable at room temperature. Reducing grain sizes to the micron scale by severe plastic deformation processes, such as equal channel angular extrusion (ECAE) or high pressure torsion, can activate grain boundary sliding to improve ductility[14, 15], but such samples fracture after about 0.2 compression strain[16]. Furthermore, magnesium products produced by severe plastic deformation processes are too small to be used industrially (i.e., in the automotive industry) and cannot be upscaled.

In this work, we report a breakthrough in the design and development of formable magnesium–polycrystalline pure magnesium can be tailored to be super-formable at room temperature by conventional processes.

## Results

**Super-formability of magnesium in cold compression or rolling.** Polycrystalline pure magnesium becomes super-formable at room temperature after it is extruded at or below 80 °C. Specimens extruded in the temperature range 150–400 °C have poor formability at room temperature. They fracture when compressed by 20–30% reduction in height, consistent with previous studies[17, 18]. However, specimens extruded at or below 80 °C do not fracture during compression at room temperature and a strain rate of $10^{-3}$ s$^{-1}$ (Fig. 1a, b). In contrast to the high work hardening of the specimens extruded at higher temperatures, these super-formable specimens exhibit no work hardening after yielding. Instead, the true stress decreases gradually with strain, implying minimal twinning and dislocation slip during the compression. Such behaviour is associated with grain boundary sliding[19] and/or dynamic recrystallization[20, 21] in magnesium alloys tested at elevated temperature. Photos in Fig. 1b show the compression test results of the specimens extruded at 80 and 400 °C. While the 400 °C extruded specimen fractures after ~20% height reduction, the 80 °C extruded specimen can be compressed

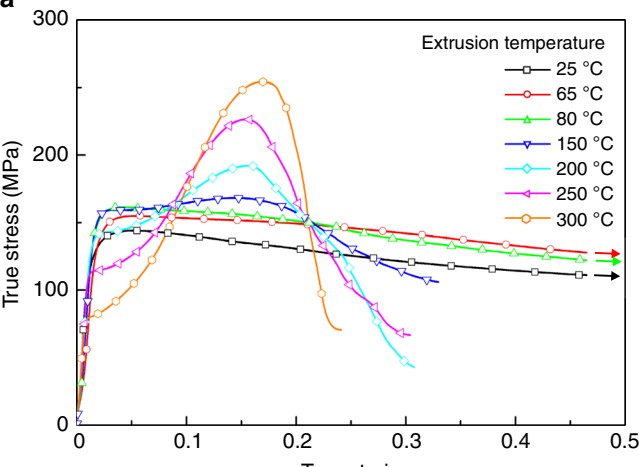

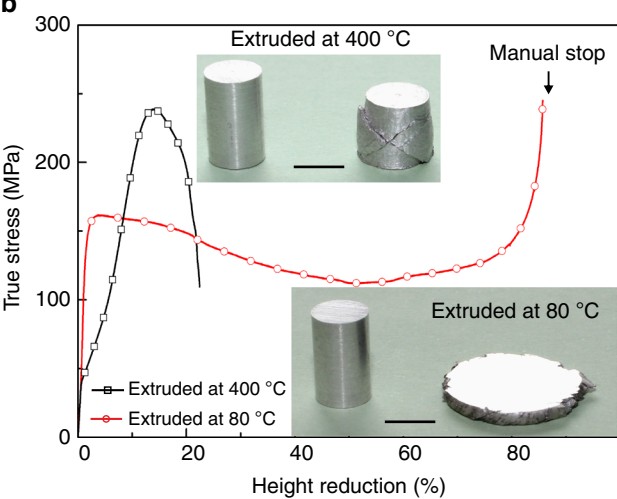

**Fig. 1** Cold compression of extruded specimens. **a** Room temperature compressive true stress-strain curves of specimens extruded **a** in the temperature range 25–300 °C. **b** Room temperature compression of specimens extruded at 80 and 400 °C. Photo insets show the specimens before and after compression test. Scale bars in photo insets indicate 5 mm

from 10 to 1.5 mm without fracture. Further height reduction from 1.5 mm is possible if the compression test continues.

As a further demonstration of the super-formability of the extruded polycrystalline pure magnesium, specimens extruded at 80 °C were also rolled at room temperature without any intermediate annealing. Their thickness was reduced continuously from 3 to 1 mm without any edge cracking. The 1 mm-thick sheet was further cold rolled to 0.5 mm, and even 0.12 mm (96% total thickness reduction equating to a true strain of 3.2). The resultant 0.12 mm-thick strip was cut into two pieces that were shaped into letters "m" and "g", as shown in Fig. 2a. This result is in distinct contrast to the ~ 30% thickness reduction (~ 0.4 true strain) obtained from conventionally processed magnesium subsequently cold rolled[7]. Even more remarkably, the cold-rolled 1 mm-thick sheet was able to be bent through 180° (Fig. 2b). Note that steel and aluminium sheet of this thickness are the common automotive body panel materials, and that it is critical for magnesium sheet to be amenable to hemming (the term given to bending through 180°) for the fabrication of automotive panels. Herein, we also demonstrated that a 0.12 mm-thick foil was readily capable of being folded twice and unfolded without any visible cracks (Fig. 2c). This is again significantly

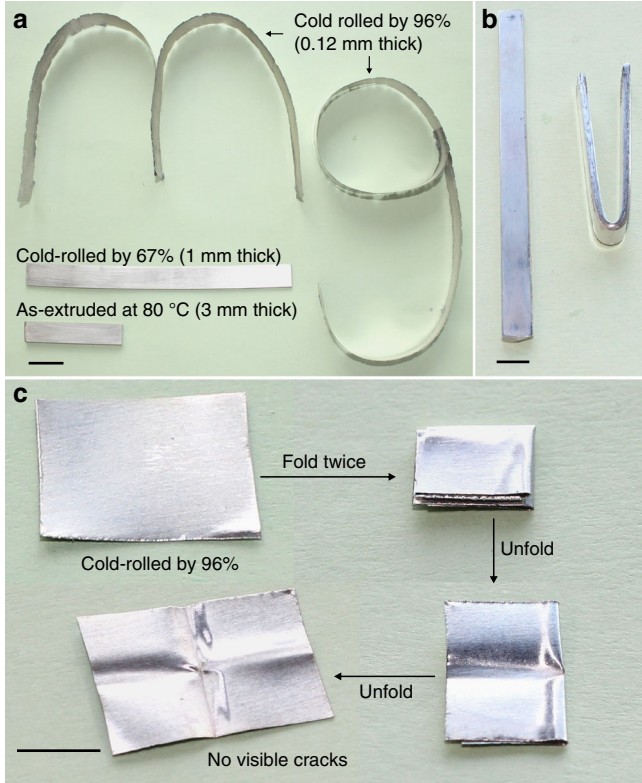

**Fig. 2** Cold rolling of extruded specimens. **a** Photo showing one 3 mm thick plate extruded at 80 °C, and after 67 and 96% cold rolling without any trimming of specimen edges along the rolling direction. The strip cold rolled by 96% was cut into two pieces that were shaped into letters "mg". **b** Photo showing a cold-rolled 1 mm strip that is bent by ~180° (spring back by ~10°) at room temperature. **c** Photos showing a cold-rolled 0.12 mm thick foil folded twice, then unfolded, without any visible cracks. Scale bars in **a**–**c** indicate 20, 3 and 5 mm, respectively

different from the traditional view that magnesium would fracture after heavy cold work or bending. Our findings demonstrate that the extruded pure magnesium remains super-formable, even after substantial plastic deformation at room temperature. It should be mentioned that pure magnesium sheet fabricated by conventional hot rolling has poor formability at room temperature. For example, for magnesium sheet produced by hot rolling at 400 °C and full annealing at 350 °C, the 1 mm-thick sheet fractured when it was bent by only ~95° (Supplementary Fig. 1a), and the 0.12 mm-thick foil cracked when it was folded once and unfolded (Supplementary Fig. 1b).

**Microstructure evolutions and deformation modes during cold deformation**. To reveal the origin of the room temperature super-formability, microstructures of specimens extruded at 400 and 80 °C were examined. Both specimens have strong basal texture (Supplementary Fig. 2) and contain predominantly equiaxed grains (Fig. 3a, f). The average grain size is ~82 and ~1.3 μm in diameter in the specimens extruded at 400 and 80 °C, respectively (Supplementary Fig. 3a, d). For the 400 °C extruded specimen after 20% cold compression or rolling, the average grain size decreases to 56–61 μm (Supplementary Fig. 3b, c), due to the presence of some twins resulting from the cold deformation (Fig. 3b, c). In contrast, there is little change in the size and shape of grains in the 80 °C extruded specimen, after even 50% cold compression or rolling (Fig. 3g, h, and Supplementary Fig. 3e, f). Viewed from the different surfaces of the cold-deformed

specimens, the average grain size is 1.1–1.2 μm. Even after the specimen is cold-rolled into a 0.12-mm thick foil, the grains remain equiaxed, and their size distribution is similar to those in the as-extruded condition (Supplementary Fig. 4). For the 80 °C extruded specimens, the basal texture becomes slightly stronger after the cold deformation (Supplementary Fig. 2e–h). Despite the large cold deformation of the specimen extruded at 80 °C, there is much less orientation deviation inside individual grains than the specimen extruded at 400 °C, as shown in grain orientation spread (GOS) maps[22] (Fig. 3d, e, i, j). The preservation of the size and shape of the grains, together with insignificant orientation deviations inside individual grains, indicate minimal intra-granular deformation in the 80 °C extruded specimen during plastic deformation at room temperature.

In order to examine the operating deformation modes, polished specimen surfaces, prepared parallel or perpendicular to the extrusion direction, were observed after cold deformation. For the 400 °C extruded specimen that was compressed by 20%, a large number of deformation twins and slip traces are observed (Fig. 4a, b). In contrast, deformation twins and slip traces are rarely detectable in the specimen extruded at 80 °C. For this specimen, the microstructure of the same area before and after cold compression was examined (Fig. 4c, d), using a quasi-in-situ method[23]. After 6% compression, a new grain, marked by a red cross, appeared at what was originally an intersection of four grains (seen from the comparison of Fig. 4c, d). Inspection of the orientation of this new grain and those surrounding it suggests that the grain was not generated by twinning of any of the neighbouring grains. This new grain (Fig. 4d) appears to have been located beneath grains 1–4 before the compression and risen to specimen surface by grain boundary sliding during the plastic deformation. Careful examination of the orientations of the surrounding grains 1–4 indicated that each of them had experienced 3–5° rotation. To further illustrate the different deformation behaviours of specimens extruded at 80 and 400 °C, strain-rate sensitivity and activation volume were calculated from strain-rate jump test data collected at room temperature (Supplementary Fig. 5). A large strain-rate sensitivity value (~0.2) was obtained for the specimen extruded at 80 °C, which was much larger than the strain-rate sensitivity value of the specimen extruded at 400 °C (~0.02). The activation volume of specimens extruded at 80 °C (~10$b^3$) was significantly smaller than that of the specimens extruded at 400 °C (~100$b^3$), where $b$ is the Burgers vector of magnesium ($3.21 \times 10^{-10}$ m). Generally, large strain-rate sensitivity values (>0.3)[24] and small activation volumes are taken as an indicator of the occurrence of grain boundary sliding as the dominant deformation mechanism[25]. The strain-rate sensitivity of the specimen extruded at 80 °C was significantly higher than that of the specimen extruded at 400 °C, indicating enhanced grain boundary sliding in plastic deformation[26, 27], whilst still lower than the value (>0.3)[25] traditionally taken as the dominant operation of grain boundary sliding. Therefore, another mechanism is also operating during room temperature plastic deformation of the specimen extruded at 80 °C.

As revealed from in Supplementary Fig. 6, dynamic recrystallisation occurred during extrusion at room temperature. Dynamic recrystallisation may also occur during compression/rolling at room temperature, either to accommodate grain boundary sliding or to act as an independent softening mechanism. For example, the new grain (Fig. 4d) has formed by dynamic recrystallisation and the strain inside this grain (Fig. 4e, f) might be generated after the recrystallisation. Supplementary Fig. 7 shows two other new grains that have also formed during the plastic deformation but look quite different from the red-crossed grain in Fig. 4d, f. These two grains have very low kernel average misorientation (KAM), and hence very low strain, relative to their surroundings. This is

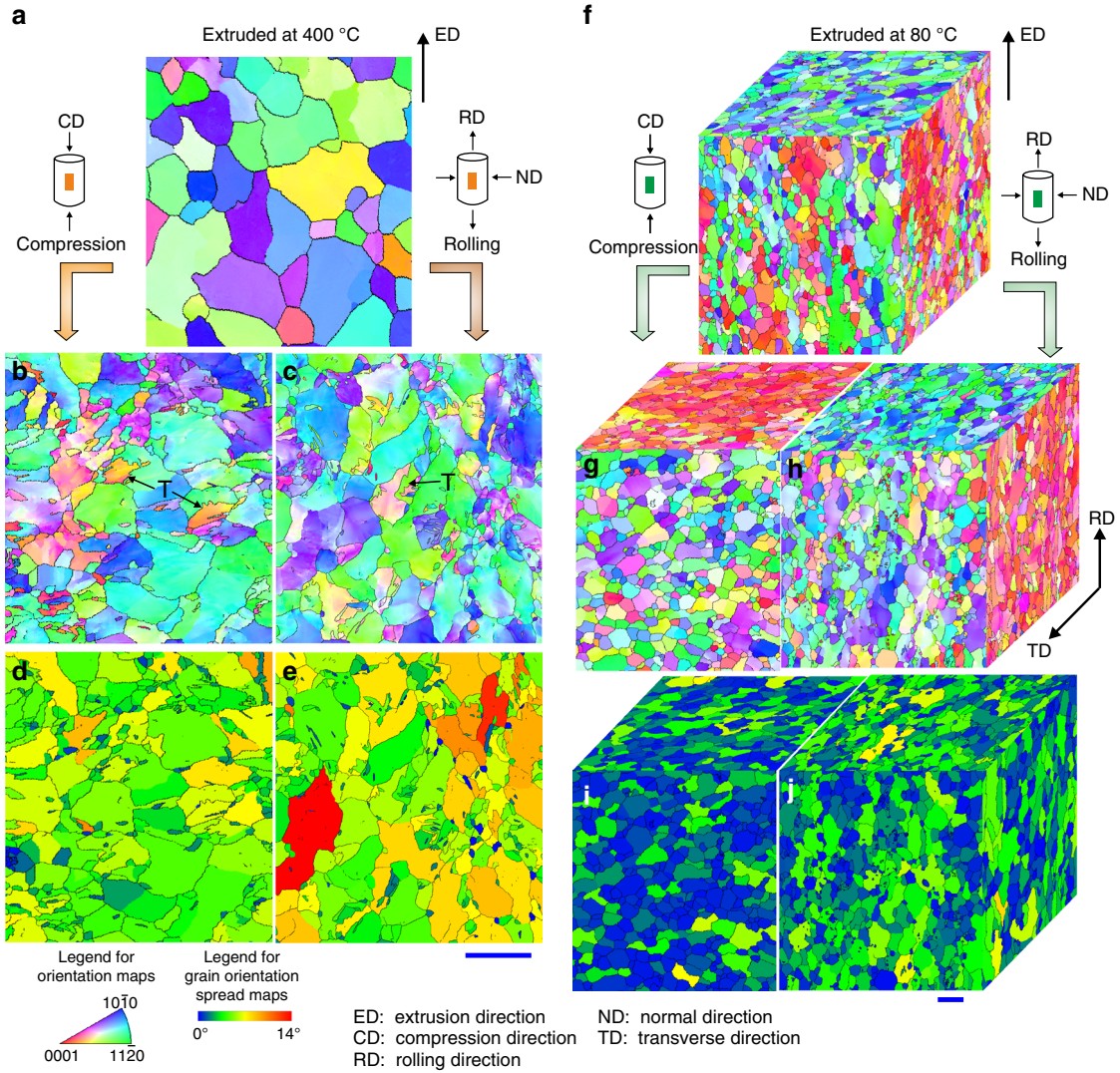

**Fig. 3** Microstructures before and after deformation. **a–e** Electron backscattered diffraction and **f–j** transmission Kikuchi diffraction maps showing microstructure of specimens extruded at **a–e** 400 °C and **f–j** 80 °C and in **a**, **f** as-extruded state, **b** after 20% and **g** 50% compression, and **c** after 20% and **h** 50% cold rolling. **d**, **e**, **i**, **j** Maps showing orientation spread in individual grains in compressed and cold-rolled specimens. Scale bars in **a–e** and **f–j** indicate 100 μm and 2 μm, respectively

exactly the characteristic expected for new grains forming by dynamic recrystallization[28]. The inter-granular mechanisms operating in tandem are consistent with the observed flow softening (Fig. 1a, b).

## Discussion

The experimental observations suggest the occurrence of significantly different deformation modes in the fine-grain specimen during cold forming, even though it also has strong basal texture. As illustrated in Fig. 5, for the coarse-grain microstructure that is usually obtained in conventionally processed magnesium and its alloys, plastic deformation occurs predominantly by dislocation slip and twinning inside individual grains. Under such circumstance, texture weakening and/or activation of more intragranular deformation modes is crucial for better formability[29, 30]. However, when the grain size is reduced to the vicinity of a micron, inter-granular sliding along grain boundaries, which is accommodated by grain rotation, and dynamic recrystallisation can be activated at room temperature. The factors that affect deformation inside grains, such as texture, dislocation slip, and twinning, thus become less significant.

Consequently, strain inside individual grains is unlikely to build to a level that will cause fracture.

Room temperature super-formability also occurs when polycrystalline pure copper (face centred cubic that has more slip systems) is cold rolled[31, 32]. In that case, the average grain size is about 28 nm, and specimens need to be prepared by a non-conventional process that cannot be used for producing bulk materials. In the case of polycrystalline pure magnesium, however, the present work demonstrates that it is not necessary to refine grain size to nano-scale in order to achieve super-formability at room temperature. Micron sized grains are sufficient, and these can be produced by conventional thermo-mechanical processes that are cost-effective, efficient and industrially scalable. Such factors are of key significance for industry in the era of light-weighting. These findings are expected to provide an important new avenue for designing and developing highly formable magnesium products.

## Methods

Commercially pure magnesium ingot (99.95%, with an average grain size of ~1.8 mm in diameter, from Amac alloys, Australia) was directly extruded at room temperature, 65, 80, 150, 200, 250, 300 and 400 °C, with an extrusion ram speed of

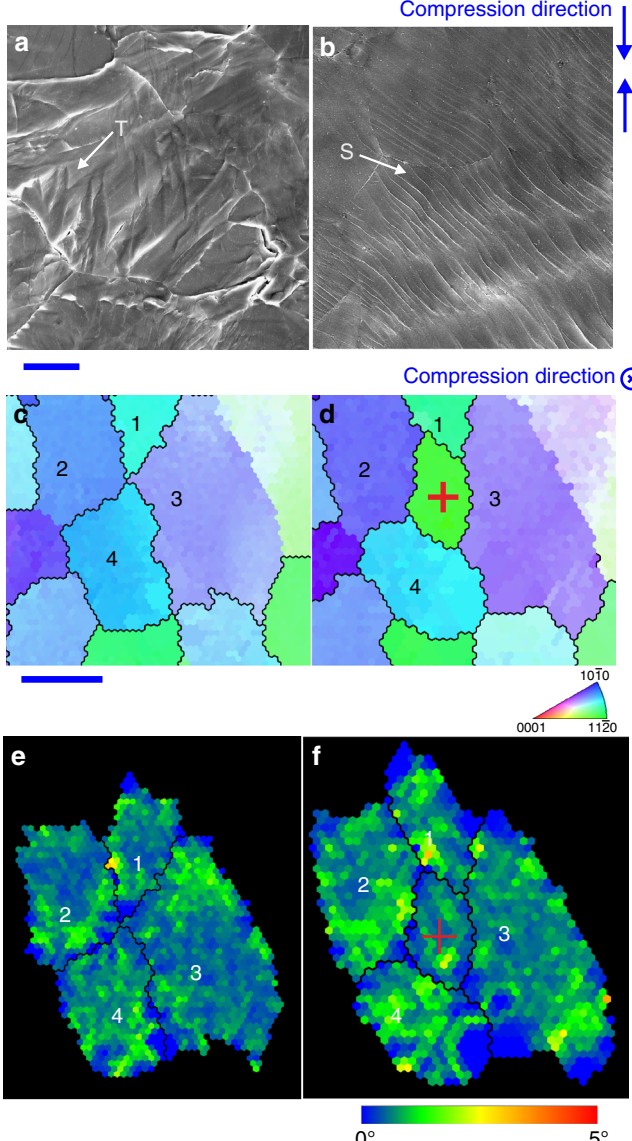

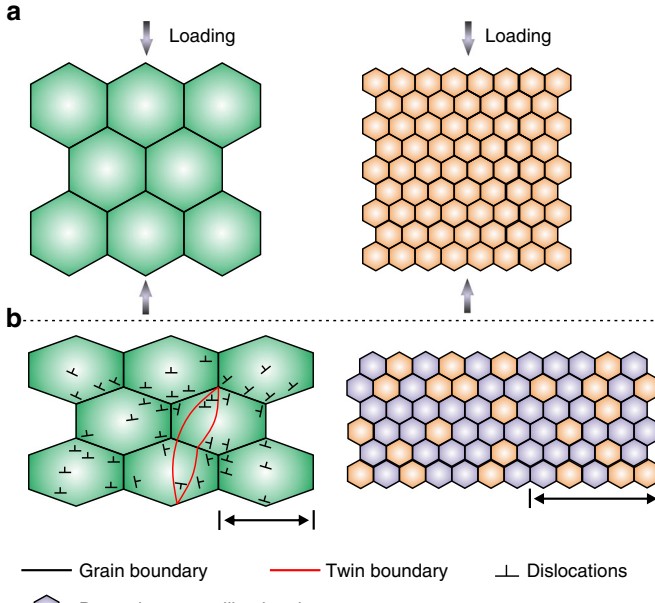

**Fig. 5** Deformation modes in coarse-grained and fine-grained specimens. Schematic diagrams showing microstructure **a** before, and **b** after compression at room temperature. The dominant deformation mechanism is intra-granular dislocation slip and twinning in a coarse-grained microstructure that is produced by 400 °C extrusion, and inter-granular mechanisms (grain boundary sliding, which is accommodated by grain rotation, and dynamic recrystallisation) in a fine-grained microstructure resulting from near room temperature extrusion. Scale bars for coarse-grained and fine-grained microstructures indicate ~80 and ~5 μm, respectively

**Fig. 4** Microstructure evolution during cold compression. Secondary electron micrographs showing **a** deformation twins (T) and **b** slip traces (S) in a specimen extruded at 400 °C and compressed by 20%. **c, d** Quasi-in-situ electron backscattered diffraction maps showing the same cross-section area of a specimen extruded at 80 °C **c** before and **d** after 6% compression in the direction perpendicular to the cross-section area. An extra grain, marked by red cross, has appeared in the area after the compression. **e, f** Kernel average misorientation maps of **c, d** qualitatively showing strain distribution in individual grains. 0° indicates low strain and 5° high strain. The red cross grain has similar kernel average misorientation. Scale bars in **a**–**c**–**f** indicate 20 μm and 500 nm, respectively

0.1 mm/s. Billets before extrusion were 35 mm in diameter, and extruded bars were 8 mm in diameter or 3 × 10 mm in cross-section, resulting in an extrusion ratio of 19:1 or 40:1, respectively. Graphite spray was applied on the billet and tools to reduce friction during extrusion. After extrusion, the as-extruded bars were immediately quenched in cold water. For cold rolling, the extruded 3 mm-thick plates were cold rolled along the extrusion direction. Thickness reduction in the cold rolling was 0.1 mm/pass, and rolling speed was 15 m/min. For comparison, benchmark sheet was produced by hot rolling at 400 °C from a 3 mm-thick slab sliced from a cast Mg ingot. The thickness reduction in the hot rolling was 20% per pass. After each rolling pass, the sheet was annealed at 400 °C for 5 min, and after the final pass, the sheet was annealed at 350 °C for 15 min.

Specimens for uniaxial compression testing were tool machined to cylinders with dimensions of 6 mm in diameter and 10 mm in height. The compression tests were performed at room temperature in an Instron 5982 machine with a crosshead speed of 0.6 mm/min. The compression direction was parallel to the extrusion direction of specimens. Two specimens were tested for each processing condition. True strain in cold rolling and compression was calculated using the equation $\varepsilon = \ln \frac{h_0}{h}$, where $h_0$ is the initial thickness/height, and $h$ is the final thickness/height of specimens after cold rolling or compression. Strain-rate sensitivity value was calculated from compression test data obtained at room temperature using the equation $m = \frac{\log(\sigma_2/\sigma_1)}{\log(\dot{\varepsilon}_2/\dot{\varepsilon}_1)}$. In the compression tests, the strain-rate was jumped among $10^{-5}$, $10^{-4}$, $10^{-3}$ and $10^{-2}$ s$^{-1}$. The activation volumes of specimens extruded at 80 and 400 °C were calculated to be ~$10b^3$ and ~$100b^3$, respectively, using the equation $V = \sqrt{3}\, kT(\frac{\partial \ln \dot{\varepsilon}}{\partial \sigma})$, where $b$ is the Burgers vector of Mg ($3.21 \times 10^{-10}$ m), $k$ is the Boltzmann constant, $T$ is the absolute temperature, $\sigma$ is the uniaxial flow stress, and $\dot{\varepsilon}$ is the strain rate of the uniaxial compression.

The three-point bending tests were performed at room temperature in an Instron 4505 machine with a punch speed of 1 mm/min. The sample size was 50 × 3 × 1 mm. The punch radius was 0.7 mm, and the distance between roller supports was 3.4 mm.

Specimens for electron backscattered diffraction (EBSD) and transmission Kikuchi diffraction (TKD) were 3 mm diameter discs with a notch parallel to the compression or rolling direction in order to ensure identification of directions in the microscope. The discs were mechanically ground to 0.12 mm thickness, then twin-jet electro-polished using a solution of 5.3 g lithium chloride, 11.2 g magnesium perchlorate, 500 ml methanol and 100 ml 2-butoxy-ethanol, at −50 °C and 0.1 A. For scanning electron microscopy (SEM), a flat surface was made parallel to the extrusion direction by grinding and polishing a round specimen to a final finish using a 50 nm diameter colloidal silica suspension. The specimen was subsequently compressed along the extrusion direction, and the flat surface of the compressed specimen was observed.

For quasi-in-situ EBSD, two specimens were machined into cylinders 6 mm in diameter and 5 mm in height. For each specimen, one surface lying perpendicular to the extrusion direction was electro-polished using a solution of 20% nitric acid and 80% methanol, at room temperature and 0.6 A. Using a Hysitron TI950 Tribo-Indentor, fiducial marks were made on the electro-polished surface of one specimen, and this surface was observed by EBSD. Then, these two specimens were stacked together, with their electro-polished surfaces face-to-face, and compressed by 6% height reduction. Silicone oil was filled between the electro-polished surfaces

to protect them from potential bonding during the compression. After compression, the specimen surface with fiducial marks was characterised again without any further polish, and the fiducial marks were used to identify the same area. This method allows observations to be made on microstructural evolution approximately that occurring inside the bulk specimen during compression.

SEM, EBSD and TKD were performed, using a FEI Quanta 3D-FEG equipped with a Pegasus Hikari EBSD detector. The data files obtained from EBSD or TKD scans were processed using TSL-OIM 6 software. The size of a grain is represented by the equivalent diameter that is calculated from the measured grain area by the equation $2\sqrt{\frac{\text{Grain area}}{\pi}}$. The average grain size in each condition is calculated by analysing all grains in a single EBSD or TKD scan, where the number of grains varies from 270 to 3000 depending on the grain size.

**Data availability**. The authors declare that the main data supporting the findings of this study are available within the article and its Supplementary Information files. Extra data are available from the corresponding author upon request.

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

## Acknowledgements

The authors are grateful for financial support from the Australian Research Council and Baoshan Iron & Steel Co., Ltd. We acknowledge the use of facilities at the Monash Centre for Electron Microscopy. Special thanks to Enrico Seemann and Daniel Curtis for their help with extrusion. N.B. is supported by Woodside Energy.

## Author contributions

Z.Z. and J.-F.N. designed the experiments, analysed the data, interpreted the results and wrote the paper. Z.Z. carried out all experiments. J.-F.N. and N.B. coordinated the two projects on wrought magnesium alloys. C.H.J.D. and N.B. contributed to the interpretation of results and writing. S.-W.X. participated the project and provided proof-reading of the manuscript.

## Additional information

**Competing interests:** The authors declare no competing financial interests.

