## [Peer Review File · Nature Communications]

Reviewers' Comments:

Reviewer #1:

Remarks to the Author:

As per my previous review I think of the present research work to be of significant scientific, ecological and economical importance to the field of magnesium alloys and their light weighting potential. There is a huge global drive towards improved fuel efficiency and reduced emissions of greenhouse gases associated with transportation, which can be met by increasing the application of lightweight Mg in automobiles. However, commercially available magnesium alloys suffer from low RT ductility and formability due to insufficient number of active deformation mechanisms and the development of strong crystallographic textures during deformation.

I would like to recapitulate that from a scientific viewpoint the work stands out among other related papers by showing that superior cold formability can be indeed achieved with the presence of a strong deformation texture. This is an extremely important and novel finding because most current attempts to improve the ductility of magnesium primarily target the texture by modifying the alloy composition, which most of the times necessitates the addition of expensive and not readily available alloying elements. A further step forward would be to broaden the scope of the formability tests applied to approach more practical conditions in terms of strain rates and strain paths. This, however, does not need to be included in the present work because in my opinion the demonstrated sheet performance (whether in compression or bending) is definitely unique and shows a big first step in the right direction.

The main reason for the exciting forming properties reported in the study is the refined homogeneous structure that led to the activation of inter-granular mechanisms (grain boundary sliding and dynamic recrystallization) instead of conventional slip and twinning within grains (intra-granular). However, the starting as-cast material had a roughly 2 mm grain size, yet it was successfully extruded between RT and 80°C, accommodating a large strain without failure and giving rise to ~1µm grain size. What are the main factors/mechanisms behind such an astonishing event?

Reviewer #2:

Remarks to the Author:

This paper addresses a very important issue: the workability or formability of Mg. The results presented in this paper are very surprising, albeit very interesting. The authors claim that after extrusion at relatively low temperatures (<80°C) with very large reduction, commercially pure Mg can be further cold rolled to very significant thickness reduction without obvious edge cracking. What is more, the final rolled ribbons can be further deformed without failure. On the contrary, those upon high temperature extrusion cannot even survive uni-axial compression to quite small strain. It is fair to say that this work is the first of its kind in the context of Mg.

The authors combine mechanical testing with detailed microstructure examination in order to

provide a plausible explanation for the extraordinary experimental observations. The stress strain curve of the material upon low-temperature extrusion shows homogeneous strain softening, while those after hot extrusion exhibit the typical sigmoidal behavior because of the activation of deformation twinning. The grain size of the low-temperature extruded billets has been refined to $\sim 1.0\mu\text{m}$. They authors believe that it is the much refined grain size that renders the "super-formable" Mg. The "super-formability" is attributed to grain boundary activities such as grain boundary sliding combined with dynamic recrystallization.

Overall this is a high-quality article and merits publication by Nature Comm. However, I strongly suggest that the authors do one more thing to improve the quality of the paper. As GB sliding is reckoned as the plastic deformation mechanism for the fine-grained Mg, it should point to quite high strain rate sensitivity and quite small activation volume associated with plastic deformation. This will render the explanation more solid and convincing, and this might be the easiest approach. Strain rate sensitivity may be evaluated by various methods such as rate jump, stress relaxation, nanoindentation at different loading rates, and so on.

Reviewer #3:

Remarks to the Author:

In this paper a method to develop super formable pure magnesium at room temperature is described. This procedure consists on extruding pure magnesium at a temperature of 80 degrees C, which results in homogeneous grain refinement down to an average value close to 1 micron. The extruded billets thus produced are shown to have remarkably good properties in compression and also excellent cold rollability and bend ability. The authors attribute these properties to a combination of dynamic recrystallization and grain boundary sliding during deformation.

The methodology utilized by the authors to fabricate super-formable magnesium is per se not new. In fact, there are many papers in the literature aiming to refine grain size in magnesium and magnesium alloys by deformation (extrusion, rolling, severe plastic deformation techniques) with the ultimate goal of enhancing strength and ductility (for example, Valiev, Nature Materials, 2004). Additionally, the mechanisms invoked to be responsible for the large ductilities observed are also well known.

The main highlight of the paper is, in my opinion, to have found the extrusion temperature (probably within a narrow temperature range) which leads to the development of the fine-grained, super formable microstructure. As such, this paper communicates optimum processing conditions, rather than a new processing methodology.

Having said that, the properties of the pure magnesium extruded at 80 degrees C are extraordinary, and therefore in my opinion this paper merits publication in Nature Communications. Pure magnesium has in general low strength but it is foreseen that low temperature extrusion could be applied to other high strength magnesium alloys in order to increase their formability.

Response to Reviewers' Comments

We would like to thank the reviewers for their comments on our manuscript. We have revised the manuscript to address these comments. We hope that the revised manuscript is now acceptable for publication in Nature Communications.

Reviewer #1

As per my previous review I think of the present research work to be of significant scientific, ecological and economical importance to the field of magnesium alloys and their light weighting potential. There is a huge global drive towards improved fuel efficiency and reduced emissions of greenhouse gases associated with transportation, which can be met by increasing the application of lightweight Mg in automobiles. However, commercially available magnesium alloys suffer from low RT ductility and formability due to insufficient number of active deformation mechanisms and the development of strong crystallographic textures during deformation.

*I would like to recapitulate that from a scientific viewpoint the work stands out among other related papers by showing that superior cold formability can be indeed achieved with the presence of a strong deformation texture. **This is an extremely important and novel finding** because most current attempts to improve the ductility of magnesium primarily target the texture by modifying the alloy composition, which most of the times necessitates the addition of expensive and not readily available alloying elements. **A further step forward would be to broaden the scope of the formability tests applied to approach more practical conditions in terms of strain rates and strain paths. This, however, does not need to be included in the present work** because in my opinion the demonstrated sheet performance (whether in compression or bending) is definitely unique and shows a big first step in the right direction.*

Response: Super-formability at room temperature and with the presence of a strong texture is indeed a breakthrough discovery. Other forms of formability tests such Erichsen and deep drawing will be performed in the near future, using much larger (industrial scale) samples and under more practical forming conditions.

*The main reason for the exciting forming properties reported in the study is the refined homogeneous structure that led to the activation of inter-granular mechanisms (grain boundary sliding and dynamic recrystallization) instead of conventional slip and twinning within grains (intra-granular). However, the starting as-cast material had a roughly 2 mm grain size, yet it was successfully extruded between RT and 80°C, accommodating a large strain without failure and giving rise to ~1µm grain size. **What are the main factors/mechanisms behind such an astonishing event?***

Response: Dynamic recrystallization is the principal mechanism that reduces grain size during extrusion at 80°C (and even at room temperature). To illustrate this point, we have now included a new figure in the revised manuscript (Extended Data Figure 6, which is also shown below). This figure reveals coarse grains in the billet before it passes through the

extrusion die, and fine grains in the other portion of the sample that had passed the extrusion die at room temperature.

Extended Data Figure 6. (a) EBSD orientation map showing the microstructure of the residual portion of an originally cylindrical billet of pure Mg that had not passed through the extrusion die (area marked red in (c)), (b) a higher magnification of the area marked by the dark rectangular frame in (a). (c) Schematic diagram showing a specimen partially extruded at room temperature. (d) TKD orientation map and corresponding (0001) pole figure showing the microstructure of the specimen, however from the extruded portion marked blue in (c). The number in the pole figure indicates the maximum texture intensity in mrd.

Reviewer #2

*This paper addresses a very important issue: the workability or formability of Mg. The results presented in this paper are very surprising, albeit very interesting. The authors claim that after extrusion at relatively low temperatures (<80°C) with very large reduction, commercially pure Mg can be further cold rolled to very significant thickness reduction without obvious edge cracking. What is more, the final rolled ribbons can be further deformed without failure. On the contrary, those upon high temperature extrusion cannot even survive uni-axial compression to quite small strain. It is fair to say that **this work is the first of its kind in the context of Mg.***

Response: We appreciate this positive comment.

The authors combine mechanical testing with detailed microstructure examination in order to provide a plausible explanation for the extraordinary experimental observations. The stress strain curve of the material upon low-temperature extrusion shows homogeneous strain softening, while those after hot extrusion exhibit the typical sigmoidal behavior because of the activation of deformation twinning. The grain size of the low-temperature extruded billets has been refined to ~1.0 μ m. They authors believe that it is the much refined grain size that renders the “super-formable” Mg. The “super-formability” is attributed to grain boundary activities such as grain boundary sliding combined with dynamic recrystallization.

Overall this is a high-quality article and merits publication by Nature Comm. However, I strongly suggest that the authors do one more thing to improve the quality of the paper. As GB sliding is reckoned as the plastic deformation mechanism for the fine-grained Mg, it should point to quite high strain rate sensitivity and quite small activation volume associated with plastic deformation. This will render the explanation more solid and convincing, and this might be the easiest approach. Strain rate sensitivity may be evaluated by various methods such as rate jump, stress relaxation, nanoindentation at different loading rates, and so on.

Response: To address the concern of the reviewer, we performed strain-rate jump tests at room temperature for specimens extruded at 80°C and 400°C. These specimens were compressed at different strain rates ranging from 10^{-5} to 10^{-2} s $^{-1}$. The test results are now shown in a newly added figure (Extended Data Figure 5, which is also shown below) in the revised manuscript. It was found that the specimen extruded at 80°C has much higher strain-rate sensitivity (0.2) and lower activation volume (~10b 3) than those extruded at 400°C (0.02 and ~100b 3), where b is the Burgers vector of magnesium (3.21×10^{-10} m). The activation volume is calculated using equation $V = \sqrt{3} kT \left(\frac{\partial \ln \dot{\epsilon}}{\partial \sigma} \right)$, k is the Boltzmann constant, T is the absolute temperature, σ is the uniaxial flow stress, and $\dot{\epsilon}$ is the strain-rate of the uniaxial compression^{R1, R2}. Generally, large strain-rate sensitivity values (> 0.3)^{R3} and small activation volumes are taken as an indicator of the occurrence of grain boundary sliding as the dominant deformation mechanism^{R4}, as suggested by the reviewer. While the strain-rate sensitivity of the specimen extruded at 80°C is much larger than that of the specimen extruded at 400°C, which indicates enhanced grain boundary sliding in the plastic deformation^{R1, R5}, it is still lower than the value (> 0.3) traditionally taken as the dominant operation of grain boundary sliding. Therefore, an other mechanism is also operating during the plastic deformation of specimens extruded at 80°C. As shown clearly in Extended Data Figure 6, dynamic recrystallization can occur at room temperature. The dynamic recrystallization may also occur during compression / rolling at room temperature, either to accommodate grain boundary sliding or to act as an independent softening mechanism. Irrespective of the details, the inter-granular mechanism permits the specimen extruded at 80°C to exhibit super-formability at room temperature.

Extended Data Figure 5. Room temperature true stress-strain curves showing strain rate jump tests of samples extruded at 80 °C and 400 °C, compressed at different strain rates in the range $10^{-5} - 10^{-2} \text{ s}^{-1}$.

References

- R1 Chen, J., Lu, L & Lu, K. Hardness and strain rate sensitivity of nanocrystalline Cu. *Scripta Mater* **54**, 1913-1918 (2006).
- R2 Frost, H. J. & Ashby, M.F. *Deformation mechanism maps, the plasticity and creep of metals and ceramics*. (Pergamon Press, 1982).
- R3 Dieter, G.E. *Mechanical Metallurgy*, 3rd edn. (McGraw-Hill, 1988).
- R4 Wang, Y. M. & Ma, E. Strain hardening, strain rate sensitivity, and ductility of nanostructured metals. *Mater Sci Eng A* **375-377**, 46-52 (2004).
- R5 Choi, H. J., Kim, Y., Shin, J. H. & Bae, D. H. Deformation behaviour of magnesium in the grain size spectrum from nano- to micrometer. *Mater Sci Eng A* **527**, 1565-1570 (2010).

Reviewer #3

In this paper a method to develop super formable pure magnesium at room temperature is described. This procedure consists on extruding pure magnesium at a temperature of 80 degrees C, which results in homogeneous grain refinement down to an average value close to 1 micron. The extruded billets thus produced are shown to have remarkably good properties in compression and also excellent cold rollability and bend ability. The authors attribute these properties to a combination of dynamic recrystallization and grain boundary sliding during deformation.

*The methodology utilized by the authors to fabricate super-formable magnesium is per se not new. In fact, there are many papers in the literature aiming to refine grain size in magnesium and magnesium alloys by deformation (extrusion, rolling, severe plastic deformation techniques) **with the ultimate goal of enhancing strength and ductility (for example, Valiev, Nature Materials, 2004).***

Additionally, the mechanisms invoked to be responsible for the large ductilities observed are also well known.

Response: In the original manuscript, we included a reference (#14) on the effects of severe plastic deformation (and hence grain refinement) on ductility. Now in the revision, we have added Valiev's paper (Nanostructuring of metals by severe plastic deformation for advanced properties. *Nature Mater* **3**, 511-516, 2004). We agree that reducing grain size to the micron scale by severe plastic deformation processes, such as equal channel angular extrusion or high pressure torsion, can activate grain boundary sliding to improve ductility, but such samples fracture after about 0.2 compression strain (References 14 and 16 in the revision). Our paper reports for the first time that the conventionally difficult-to-form Mg can become super-formable at room temperature even when it is manufactured by conventional thermomechanical processes that are cost-effective, efficient, and industrially scalable.

The main highlight of the paper is, in my opinion, to have found the extrusion temperature (probably within a narrow temperature range) which leads to the development of the fine-grained, super formable microstructure. As such, this paper communicates optimum processing conditions, rather than a new processing methodology.

*Having said that, **the properties of the pure magnesium extruded at 80 degrees C are extraordinary**, and therefore in my opinion this paper merits publication in Nature Communications. Pure magnesium has in general low strength but it is foreseen that low temperature extrusion could be applied to other high strength magnesium alloys in order to increase their formability.*

Response: We appreciate the positive comment of the reviewer.

Reviewers' Comments:

Reviewer #1:

Remarks to the Author:

The authors have satisfactorily addressed all comments. The additional data is helpful and significantly strengthens the arguments being made. The paper is of high scientific quality and merits publication in Nature Communications.

Reviewer #2:

Remarks to the Author:

The authors have addressed all the comments and have modified their manuscript. According to this reviewer, the manuscript can now be accepted for publication by NatComm.

Response to Reviewers' Comments

Reviewer #1 (Remarks to the Author):

The authors have satisfactorily addressed all comments. The additional data is helpful and significantly strengthens the arguments being made. The paper is of high scientific quality and merits publication in Nature Communications.

Response: We appreciate this positive comment.

Reviewer #2 (Remarks to the Author):

The authors have addressed all the comments and have modified their manuscript. According to this reviewer, the manuscript can now be accepted for publication by NatComm.

Response: We appreciate this positive comment.